# The Shu complex prevents mutagenesis and cytotoxicity of single-strand specific alkylation lesions

Braulio Bonilla[1], Alexander J Brown[2], Sarah R Hengel[1], Kyle S Rapchak[1], Debra Mitchell[2], Catherine A Pressimone[1], Adeola A Fagunloye[1], Thong T Luong[1], Reagan A Russell[3], Rudri K Vyas[2], Tony M Mertz[2], Hani S Zaher[4], Nima Mosammaparast[5], Ewa P Malc[6], Piotr A Mieczkowski[6], Steven A Roberts[2]*, Kara A Bernstein[1]*

[1]Pharmacology and Chemical Biology, University of Pittsburgh School of Medicine, Pittsburgh, United States; [2]Molecular Biosciences and Center for Reproductive Biology, Washington State University, Pullman, United States; [3]University of Pittsburgh School of Medicine, Pittsburgh, United States; [4]Biology, Washington University in St Louis, St. Louis, United States; [5]Washington University in St Louis, St Louis, United States; [6]Genetics, University of North Carolina Chapel Hill, Chapel Hill, United States

**Abstract** Three-methyl cytosine (3meC) are toxic DNA lesions, blocking base pairing. Bacteria and humans express members of the AlkB enzymes family, which directly remove 3meC. However, other organisms, including budding yeast, lack this class of enzymes. It remains an unanswered evolutionary question as to how yeast repairs 3meC, particularly in single-stranded DNA. The yeast Shu complex, a conserved homologous recombination factor, aids in preventing replication-associated mutagenesis from DNA base damaging agents such as methyl methanesulfonate (MMS). We found that MMS-treated Shu complex-deficient cells exhibit a genome-wide increase in A:T and G:C substitutions mutations. The G:C substitutions displayed transcriptional and replicational asymmetries consistent with mutations resulting from 3meC. Ectopic expression of a human AlkB homolog in Shu-deficient yeast rescues MMS-induced growth defects and increased mutagenesis. Thus, our work identifies a novel homologous recombination-based mechanism mediated by the Shu complex for coping with alkylation adducts.

*For correspondence:
steven.roberts2@wsu.edu (SAR);
karab@pitt.edu (KAB)

Competing interest: The authors declare that no competing interests exist.

## Editor's evaluation

The study supports the existence of a new pathway for the removal of an important DNA lesion, 3meC in single stranded DNA during replication, that seems essential in yeast, but likely contributes in other organisms, and helps clarify the distinctive role of homologous recombination in DSB repair and post-replicative repair. The paper is of interest to an audience of DNA repair and cancer biologists because it seeks to refine the mechanism by which cells respond to DNA damage. This is an important topic, and the results are consistent with previous work in the field. New claims could change our understanding of DNA repair with implications for mutagenesis and cancer therapy.

## Introduction

Alkylating agents such as methyl methanesulfonate (MMS) induce a diverse set of base lesions that are recognized and repaired by the base excision repair (BER) pathway or direct repair enzymes such

as the AlkB family (*Fu et al., 2012*; *Yi and He, 2013*). However, when these lesions, or its repair inter-mediates, are encountered by a replisome, replication fork stalling can occur (*Shrivastav et al., 2010*; *Sobol et al., 2003*). In this scenario, DNA base damage is preferentially bypassed using homologous recombination (HR) or translesion synthesis (TLS), postponing its repair but allowing replication to be completed in a timely fashion. These pathways are often referred to as post-replication repair (PRR) or DNA damage tolerance (DDT) and are best described in the budding yeast *Saccharomyces cerevisiae* (*Arbel et al., 2020*; *Boiteux and Jinks-Robertson, 2013*; *Ulrich, 2007*).

In yeast, HR-mediated PRR is an error-free pathway and is dependent on the polyubiquitination of PCNA by the Mms2-Rad5-Ubc13 complex (*Arbel et al., 2020*). Lesion bypass is achieved by Rad51 filament formation, and recombination between sister chromatids to fill the single-stranded DNA (ssDNA) gaps originated from the stalling of replicative polymerases. While HR can bypass these lesions in an error-free manner, TLS can also bypass this damage. However, TLS may lead to mutations and is often referred to as error-prone lesion bypass. In yeast, TLS serves as an alternative pathway to error-free lesion bypass, as disruption of genes involved in the error-free PRR pathway leads to an increase in mutations that is dependent on TLS (*Broomfield et al., 1998*; *Huang et al., 2003*; *Swanson et al., 1999*).

The study of the error-free PRR pathway is challenging due to the diverse roles of HR proteins in additional processes, such as double-strand break (DSB) repair. The Shu complex is an evolutionarily conserved HR factor that we recently discovered to function in strand-specific DDT during replication and thus provides an excellent model to specifically study HR proteins in the context of PRR (*Godin et al., 2015*; *Godin et al., 2016c*; *Martino and Bernstein, 2016*; *Rosenbaum et al., 2019*). In *S. cerevisiae*, the Shu complex is a heterotetramer formed by the SWIM domain-containing protein, Shu2, and the Rad51 paralogs, Csm2, Psy3, and Shu1. The Shu complex promotes Rad51 filament formation, a key step for HR (*Godin et al., 2016c*). Consistent with their role in DSB repair, most HR genes deletions lead to increased sensitivity to DSB-inducing agents. However, this is not the case with the Shu complex, as its mutants are primarily sensitive to the alkylating agent MMS, but not to the DSB-inducing agents such as ionizing radiation (*Godin et al., 2016b*; *Shor et al., 2005*). This makes the Shu complex attractive to dissect the role of HR in the tolerance of replication-associated DNA damage.

Previous studies from our group and others demonstrated that the Shu complex operates in the error-free branch of the PRR to tolerate DNA damage from MMS-induced lesions (*Ball et al., 2009*; *Godin et al., 2013*). Despite these findings, it has remained unknown which MMS-induced lesions, or repair intermediates, the Shu complex is important for. Our previous results uncovered genetic inter-actions between factors of the BER pathway and the Shu complex upon MMS treatment (*Godin et al., 2016b*). Notably, cells lacking the BER purinic/apyrimidinic site (AP) endonucleases and AP lyases that process AP sites exhibit exquisite sensitivity toward alkylating agents and display a three order of magnitude increase in mutations rates when the Shu complex is disrupted. We recently demonstrated that the Shu complex is important for the tolerance of AP sites, also referred to as abasic sites (*Rosenbaum et al., 2019*). However, it remains an outstanding question as to whether the Shu complex may recognize other MMS-induced fork blocking lesions.

Here, we addressed whether the Shu complex function is specific for AP sites or if it is important for the recognition of a broader range of DNA lesions. To do this, we performed whole genome sequencing of Shu mutant cells (i.e. *csm2Δ*) that had been chronically exposed to MMS. In turn, this allowed us to obtain an unbiased spectrum of mutations the Shu complex helps to mitigate. We determined that the Shu complex prevents mutations at both A:T and C:G base pairs. However, tran-scriptional and replicative asymmetries in C:G mutations suggest a novel role for the Shu complex in the tolerance of three-methyl cytosine (3meC), in addition to AP sites. Importantly, unlike bacteria and human cells that have an enzyme that directly repairs 3meC, this family of enzymes is absent in *S. cerevisiae* and therefore it has remained unknown how 3meC are repaired in yeast (*Sedgwick et al., 2007*). Our data suggest that the Shu complex is important for 3meC tolerance. Indeed, expression of human *ALKBH2*, responsible for 3meC repair, specifically rescues the MMS sensitivity of Shu complex mutant cells and alleviates their MMS-induced mutagenesis. In contrast to Shu complex mutants, we observe that *ALKBH2* expression very weakly rescues the MMS sensitivity associated with HR dele-tion strains. Altogether, our findings reveal a previously unappreciated role for the Shu complex in mediating damage tolerance of 3meC in ssDNA, finally uncovering how yeast tolerate these highly

toxic lesions. Additionally, the ability of the Shu complex to facilitate tolerance of 3meC DNA damage through homology-directly bypass likely highlights an important repair pathway for these lesions even in organisms that express AlkB homologs.

## Results

### Unbiased genome-wide analysis of mutation patterns suggests that the Shu complex function in the error-free bypass of specific MMS-induced lesions

To determine the identity of MMS-induced lesions that the Shu complex helps the replication fork to bypass, we chronically exposed Shu complex-deficient cells to MMS. In particular, wild-type (WT) or *csm2Δ* cells were plated on MMS-containing medium and then individual colonies were transferred every 2 days onto fresh medium containing 0.008 % MMS, for a total of 10 passages. We then extracted genomic DNA from these colonies and performed whole genome sequencing to measure mutation frequencies. Consistent with previous findings, *csm2Δ* cells accumulated a median 12.7-fold more mutations than WT upon MMS treatment (*Figure 1A*; *Godin et al., 2013*; *Godin et al., 2016b*). Base substitutions, insertion/deletion mutations, and complex mutations (consisting of two or more mutations separated by 10 bp or less) were all elevated (*Figure 1—figure supplement 1*), with base substitutions constituting approximately 95 % of the total mutations in both the MMS-treated WT and *csm2Δ* yeast. MMS-induced mutations were largely randomly distributed across yeast genomes (*Supplementary file 1c* and *Figure 1—figure supplement 2*). To infer which DNA lesions caused the mutations, we analyzed the substitution patterns considering the MMS-induced lesion profile (*Beranek, 1990*; *Sikora et al., 2010*; *Wyatt and Pittman, 2006*). 3meA is the most common MMS-induced lesion at A:T base pairs in dsDNA and can itself be mutagenic or converted to a mutagenic AP site by the Mag1 glycosylase or by spontaneous depurination (*Figure 1B and C*; *Shrivastav et al., 2010*). When these AP sites are encountered by a replicative polymerase, they can lead to its stalling. TLS activity on 3meA-derived AP sites leads to an A-> G and A-> T substitution pattern (*Figure 1C*) due to the tendency of Rev1 or the replicative polymerase δ incorporating a C or A on AP sites, respectively (*Figure 1C*; *Chan et al., 2013*; *Haracska et al., 2001*; *Hoopes*

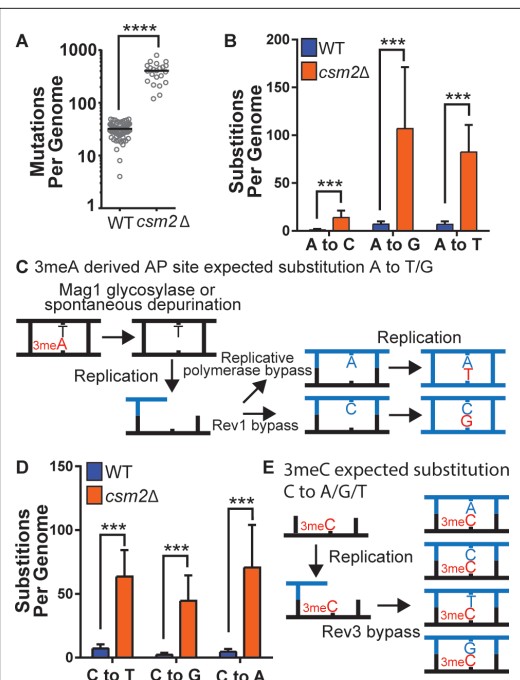

**Figure 1.** *csm2Δ* cells chronically exposed to methyl methanesulfonate (MMS) exhibit substitution patterns consistent with translesion synthesis (TLS) activity bypassing abasic (AP) sites and three-methyl cytosine (3meC). WT and *csm2Δ* cells were chronically exposed to MMS. WT and *csm2Δ* cells were chronically exposed to 0.008 % MMS by plating individual colonies onto rich medium containing MMS, after 2 days of growth, the colonies were plated onto fresh medium containing MMS for 10 passages. DNA was extracted from 75 WT and 22 *csm2Δ* clonal isolates and deep sequenced. (**A**) The number of mutations per genome for WT or *csm2Δ* cells chronically MMS-exposed. The horizontal bar indicates the median value for each genotype. **\*\*\*\*** indicates p < 0.0001 by Mann-Whitney test. (**B**) The average number of each type of A:T substitution per genome in MMS-treated WT and *csm2Δ* cells. Error bars indicate standard deviation among the samples in each group. **\*\*\*** indicates p < 0.0001 comparing the number of A:T substitutions per genome in WT and *csm2Δ* yeast by t-test. (**C**) Schematic of how 3meA-derived AP sites result in A to T/G mutations. 3meA is removed by the Mag1 glycosylase or it undergoes spontaneous depurination resulting in an AP site. During DNA replication, the replicative polymerase bypasses the AP site resulting in a T mutation, alternatively, Rev1 bypasses the AP site resulting in a G mutation. (**D**) The average number of each type of G:C substitution per genome in MMS-treated WT and *csm2Δ* cells. Error bars indicate standard deviation among the samples in each group. **\*\*\*** indicates p < 0.0001 comparing the number of G:C substitutions per genome in WT and *csm2Δ* yeast by t-test. (**E**) Schematic of how 3meC results in C to A/G/T base substitutions. 3meC occurs primarily in ssDNA and during replication,

*Figure 1 continued on next page*

*Figure 1 continued*

Rev3 mediated bypass results in incorporation of A, T, G, or C nucleotides.

The online version of this article includes the following figure supplement(s) for figure 1:

**Figure supplement 1.** Density of methyl methanesulfonate (MMS)-induced mutations in diploid wild-type (WT) and *csm2Δ/csm2Δ* yeast.

**Figure supplement 2.** Location of methyl methanesulfonate (MMS)-induced mutations on Chr.5 of independently sequenced isolates of diploid wild-type (WT) and csm2Δ/csm2Δ yeast.

**Figure supplement 3.** Spectra of methyl methanesulfonate (MMS)-induced mutations in the *CAN1* gene in *csm2Δ*, *mag1Δ*, and *mag1Δ csm2Δ* yeast.

**Figure supplement 4.** Schematic of how 7meG-derived abasic (AP) sites lead to G to T as the main substitution pattern.

*et al., 2017*). Consequently, most substitutions at A:T bases in MMS-treated WT yeast A to G and A to T (*Figure 1B*). Previous analyses of MMS-treated *mag1Δ* yeast revealed that deletion of the glycosylase responsible for initiating BER of 3meA resulted in a mutation spectrum also predominantly composed of A to G and A to T substitutions (*Mao et al., 2017*), indicating that these two mutation types are likely directly induced by 3meA as opposed to 3meA-derived AP sites. MMS-treated *csm2Δ* cells (*Figure 1B*) displayed elevated A-> G transitions and A-> T transversions, consistent with a function for the Shu complex in the bypass of 3meA or 3meA-derived AP sites, as we recently demonstrated (*Godin et al., 2016b*; *Rosenbaum et al., 2019*). MMS-induced mutations selected in the *CAN1* gene in *mag1Δ* yeast primarily occur at A:T bases (similar to MMS-induced mutations previously identified by whole genome sequencing of *mag1Δ* yeast), indicative of 3meA-induced muta-

tions, while MMS-induced *CAN1* mutations in *csm2Δ* yeast are primarily caused by C or G substitutions (*Figure 1—figure supplement 3*). However, the mutation spectrum from *mag1Δ csm2Δ* double mutant yeast mirrors that from *mag1Δ* yeast despite a previously determined higher MMS-induced mutation rate for *mag1Δ csm2Δ* than *mag1*-deficient yeast (*Godin et al., 2016b*). This spectrum, therefore, indicates that elevated frequency of A:T substitutions in *mag1Δ csm2Δ* yeast compared to *csm2Δ* yeast is due to higher amounts of 3meA present (due to the lack of Mag1) that are normally bypassed with the help of the Shu complex. 7meG is the most common MMS-induced DNA adduct at G:C base pairs and is not itself mutagenic (*Shrivastav et al., 2010*). However, it is eliminated by spontaneous depurination, or Mag1 excision, which both lead to AP sites (*Bjørås et al., 1995*; *Shrivastav et al., 2010*; *Wyatt et al., 1999*). When AP sites are generated from guanines, their bypass by Rev1 is error-free because it incorporates a C across the missing base. Alternatively, bypass by the replicative polymerase results in the incorporation of A across the missing base, which leads to a G to T transversion (*Figure 1—figure supplement 4*). Interestingly, the MMS-induced mutation profile at G:C base pairs in WT yeast not only includes G-> T mutations but also high levels of G-> A and G-> C mutations (*Figure 1D*), suggesting that 7meG-derived AP sites are unlikely to cause these mutations. Deletion of *CSM2* results in a similar spectrum of MMS-induced G:C substitutions compared to that observed in WT yeast, but at elevated frequency, indicating that the Shu complex mediates tolerance of the same MMS-induced lesion that contributes to G:C base pair mutagenesis in WT cells, possibly N3-methylcytosine (3meC). 3meC is a mutagenic DNA adduct predominantly induced by MMS at ssDNA and can block the replicative polymerase (*Shrivastav et al., 2010*). Rev3-mediated TLS bypass of 3meC leads to the incorporation of a random nucleotide across the lesion, leading to a mutational pattern consistent with the one we observed in WT and *csm2Δ* yeast (*Figure 1E*; *Saini et al., 2020*; *Yang et al., 2010*), indicating that the Shu complex may also bypass this ssDNA-specific alkylation lesion.

DNA lesions that arise at ssDNA often result in mutation patterns that exhibit strand bias near origins of replication and at highly transcribed genes (*Roberts et al., 2012*; *Saini and Gordenin, 2020*). Additionally, lesions subject to preferential repair of the transcribed DNA strand by transcription-coupled nucleotide excision repair (TC-NER) can also display transcriptional asymmetry (i.e. having higher mutation densities on the non-transcribed strand compared to the transcribed strand). Therefore, we evaluated whether mutations from our whole genome sequenced yeast displayed either transcriptional or replicative asymmetry (*Figure 2*). A:T mutations in both MMS-treated WT and *csm2Δ* yeast showed very little replicative asymmetry, but displayed transcriptional asymmetry, similar to that seen in MMS-treated *mag1Δ* yeast (*Mao et al., 2017*). The transcriptional asymmetry of A:T mutations in MMS-treated *mag1Δ* yeast results from TC-NER activity on unrepaired 3meA, indicating that the Shu complex may help prevent mutations associated with 3meA (*Beranek, 1990*; *Sikora et al., 2010*;

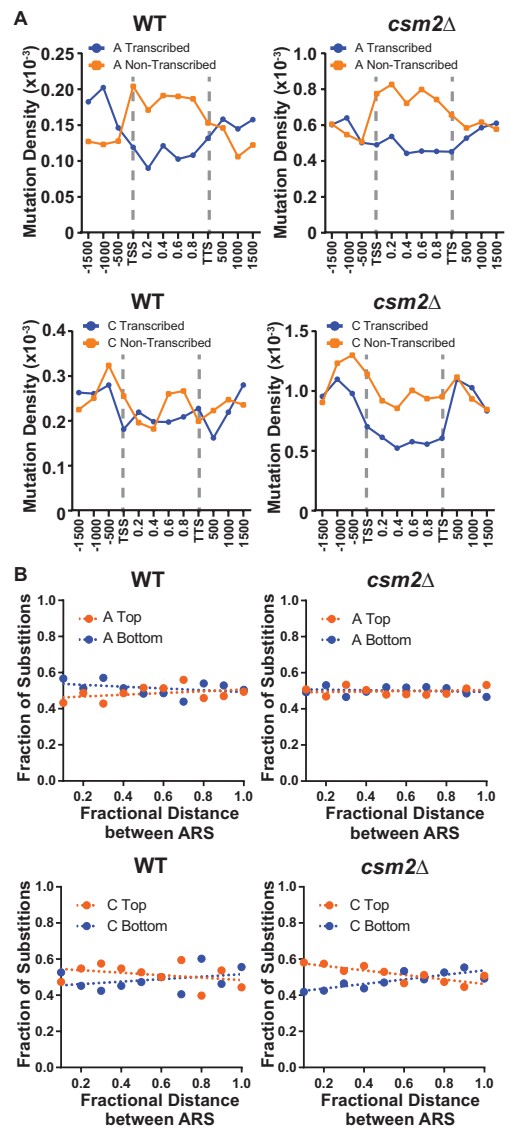

**Figure 2.** Transcriptional and replicative strand biases of methyl methanesulfonate (MMS)-induced substitutions. (**A**) The density of A mutations (i.e. the fraction of A bases mutated) or C mutations on the transcribed (blue) and non-transcribed (orange) strand across yeast transcripts in wild-type (WT) and *csm2Δ* cells. Transcript regions were broken into fractional bins of 0.2 of the transcript length and the density of A or C mutations determined per bin. Three additional bins of 500 bp each were also included upstream of the transcription start site (TSS) and downstream of the transcription termination site (TTS). (**B**) The fraction of A and C mutations occurring on the top (orange) and bottom (blue) strands across replication units in the genomes of WT and *csm2Δ* cells. Replication units were broken into 0.1 fractional bins between neighboring origins of replication and the fraction of A mutations or C mutations associated with each strand were calculated per bin. The fraction of mutations for each

*Figure 2 continued on next page*

*Figure 2 continued*

strand across the replication unit was fitted with linear regression lines (dashed lines).

The online version of this article includes the following figure supplement(s) for figure 2:

**Figure supplement 1.** Effects of transcription-coupled nucleotide excision repair (TC-NER) deficiency on methyl methanesulfonate (MMS)-induced mutations in *csm2Δ* yeast.

**Figure supplement 1—source data 1.** Numerical data corresponding to the graph in panel A.

---

*Wyatt and Pittman, 2006*). However, increased mutagenesis of the minor ssDNA-specific lesion $N^1$-methyl adenine (1meA) would also be expected to produce higher mutation densities on the non-transcribed strand. Therefore, we cannot exclude the possibility that 1meA contributes to the increased MMS-induced mutation rate in *csm2Δ* cells. G:C mutations in MMS-treated WT yeast displayed little to no transcriptional or replicative strand asymmetry, while the same mutations in MMS-exposed *csm2Δ* cells exhibited both types of strand bias (*Figure 2*). The transcriptional asymmetry displayed higher densities of C mutations on the non-transcribed strand, consistent with either TC-NER removal of 3meC or preferential formation of 3meCs on the transiently ssDNA non-transcribed strand, instead of an asymmetry derived for TC-NER of 7meG. To determine if the transcriptional asymmetry, of the G:C mutations, is due to increased 3meC on the non-transcribed strand or the result TC-NER activity, we measured the frequency and spectra of MMS-induced mutations selected in the *CAN1* gene in *csm2Δ* and *csm2Δ rad26Δ* yeast. Co-deletion of the essential TC-NER factor *RAD26* with *CSM2* failed to significantly increase MMS-induced *CAN1* mutation rates above that of *csm2Δ* alone (*Figure 2—figure supplement 1A*). Analysis of the mutation spectra in *CAN1* indicated a small transcriptional strand bias in C:G bases in the *csm2Δ* yeast, recapitulating the transcriptional asymmetry observed in our whole genome sequencing (*Figure 2—figure supplement 1B*). However, this transcriptional asymmetry remained in the spectra of the MMS-treated *csm2Δ rad26Δ* yeast, suggesting that the observed strand bias is likely the result of increased formation of 3meC on the non-transcribed strand and not due to the effects of TC-NER. We additionally note that transcriptional asymmetry of the 3meC in the whole genome sequencing of MMS-treated *csm2Δ* extends up to 167 bp upstream of the transcriptional start site. This may suggest that either TC-NER can function

within this region or some ssDNA is assessable on the non-transcribed strand due to the binding of TFIIH to the promoter region upstream of the TSS. Likewise, replicative asymmetry indicated elevated C mutations associated with the lagging strand template across the yeast genome, suggesting that 3meC lesions formed during lagging strand synthesis are bypassed in an error-free manner in Shu proficient yeast.

## *ALKBH2* expression rescues the MMS-induced phenotypes of Shu complex-deficient cells

Based on the analysis of the mutation patterns, we hypothesized that the Shu complex promotes the error-free bypass of 3meC. 3meC can be repaired by the AlkB family of Fe(II)/α-ketoglutarate-dependent dioxygenases (*Fedeles et al., 2015*). This family is conserved from bacteria to humans; however, no yeast homolog has been found. Therefore, as of yet, it remains unknown how yeast tolerates and repairs the highly toxic 3meC. In humans, there are nine AlkB homologs with ALKBH2 and ALKBH3 are responsible for 3meC DNA repair (*Fedeles et al., 2015*; *Yi and He, 2013*). We, therefore, reasoned that ectopic expression of AlkB homologs would rescue the MMS-induced phenotypes observed in Shu complex-deficient cells. To test this, we took advantage of the lack of an AlkB homolog in yeast by ectopically expressing human AlkB homologs, *ALKBH2*, or *ALKBH3*, in *csm2Δ* cells and analyzed their effect on MMS sensitivity. *ALKBH2* and *ALKBH3* were expressed under the constitutive GAP promoter using a low-copy CEN plasmid. We find that both *ALKBH2* and *ALKBH3* expression leads to a partial rescue of the growth defects observed in MMS-exposed *csm2Δ* cells, with *ALKBH2* showing a stronger rescue (*Figure 3A*). Therefore, we focused on *ALKBH2* for the remainder of the experiments. As expected, *ALKBH2* expression very mildly rescues the MMS sensitivity of WT cells (*Figure 3A*). The growth defect rescue observed in MMS-treated *csm2Δ* cells depends on ALKBH2's enzymatic activity since the expression of an *ALKBH2* catalytic dead mutant (ALKBH2-V101R,F120E herein referred to as alkbh2-CD) (*Monsen et al., 2010*) does not rescue *csm2Δ* cell viability (*Figure 3B*). We confirmed that the inability of alkbh2-CD to rescue *csm2Δ* cells was not due to lower protein expression (*Figure 3—figure supplement 1*). These findings are not specific to *csm2Δ* as *ALKBH2* rescues the MMS sensitivity of the other Shu complex members to the same extent (*Figure 3C*). The effect of *ALKBH2* expression on *csm2Δ* cell viability is specific for MMS damage as *ALKBH2* expression does not rescue the known growth defect observed in UV-treated *csm2Δ rev3Δ* double mutant cells (*Xu et al., 2013*; *Figure 3D*).

Next, we analyzed the effect of *ALKBH2* expression on *csm2Δ* cells acutely exposed to MMS. To do this, we treated *ALKBH2*-expressing WT or *csm2Δ* cultures with 0.1 % MMS for 30 min. We then assessed cell survival by counting viable colonies after 2 days of growth. We observe that *ALKBH2* expression leads to a dose-dependent increase in the survival of *csm2Δ* cells, whereas in WT cells, survival is only mildly rescued (*Figure 3E*).

Since 3meC only occurs at ssDNA, which is a physiological intermediate of DNA replication, we reasoned that *ALKBH2* expression would preferentially rescue the survival of cells that are progressing through S-phase and therefore more vulnerable to 3meC-induced toxicity. To test this, we compared the survival of *ALKBH2*-expressing *csm2Δ* cells arrested in G1 in MMS-containing media with *ALKBH2*-expressing *csm2Δ* cells progressing through S-phase. In agreement with our rationale, *ALKBH2* was found to rescue the survival phenotype only for cells progressing in the S-phase of the cell cycle, which is when an HR-dependent error tolerance occurs (*Figure 3F*). This is consistent with previous data in human cells suggesting that *ALKBH2* functions preferentially in S-phase (*Gilljam et al., 2009*).

## *Alkbh2* expression alleviates the MMS-induced mutations observed in Shu complex mutant, *Csm2Δ*

Since 3meC is a mutagenic lesion, we asked whether *ALKBH2* expression would alleviate the observed mutational load of MMS-exposed *csm2Δ* cells. To do this, we utilized the *CAN1* reporter assay. The *CAN1* gene encodes for an arginine permease, therefore when cells are exposed to the toxic arginine analog canavanine, only cells that have acquired mutations in the *CAN1* gene grow. *ALKBH2* expression decreased the mutation rates in MMS-exposed *csm2Δ* cells (*Figure 4A*; p = 0.0092). This decreased mutation rate is specific for alkylation lesions as *ALKBH2* expression fails to rescue the increase in spontaneous mutations observed in a Shu complex mutant, *csm2Δ* cells (*Figure 4A*; p = 0.1327). Furthermore, the rescue of the *csm2Δ* mutation rate by *ALKBH2* depends on *ALKBH2*

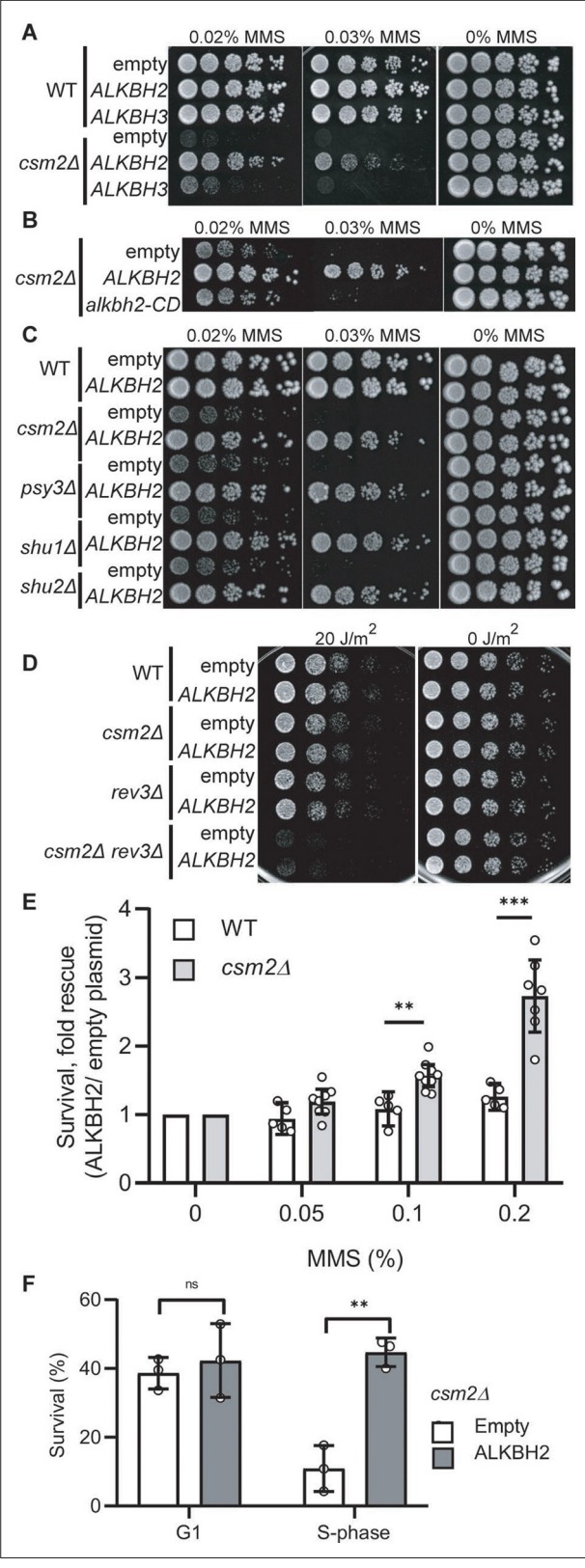

**Figure 3.** Expression of human *ALKBH2* rescues the methyl methanesulfonate (MMS) sensitivity of *csm2Δ* cells. (**A**) *csm2Δ* cells expressing *ALKBH2* exhibit decreased MMS sensitivity. Fivefold serial dilution of wild-type (WT) or *csm2Δ* cells transformed with an empty plasmid, a plasmid expressing *ALKBH2* or a plasmid expressing *ALKBH3* onto rich YPD medium or YPD medium containing the indicated MMS concentration were incubated for 2 days at

*Figure 3 continued on next page*

*Figure 3 continued*

30 °C prior to being photographed. (**B**) The enzymatic activity of ALKBH2 is required for the rescue of the MMS sensitivity of *csm2Δ* cells. *csm2Δ* cells transformed with an empty plasmid, a plasmid expressing *ALKBH2* or a plasmid expressing a catalytic dead *ALKBH2* mutant were diluted and plated as described in (**A**) and incubated for 3 days at 30 °C prior to being photographed. (**C**) *ALKBH2* expression rescues the MMS sensitivity of cells with deletions of each Shu complex gene. WT, *csm2Δ*, *psy3Δ*, *shu1Δ*, or *shu2Δ* cells transformed with an empty plasmid or a plasmid expressing *ALKBH2* were fivefold serially diluted, plated, and analyzed as described in (**B**). (**D**) Expression of *ALKBH2* does not rescue the increased ultraviolet (UV) sensitivity observed in *csm2Δ rev3Δ* double mutants. Fivefold serial dilution of WT, *csm2Δ*, *rev3Δ*, or *rev3Δ csm2Δ* cells were transformed with an empty plasmid or a plasmid expressing *ALKBH2* and fivefold serially diluted onto rich YPD or rich YPD medium exposed to 20 J/m$^2$ UV, and incubated for 2 days at 30 °C prior to being photographed. An untreated plate (0 J/m$^2$) serves as a loading control. (**E**) *csm2Δ* cells expressing *ALKBH2* exhibit increased survival after acute MMS treatment. YPD liquid cultures of WT or *csm2Δ* cells transformed with an empty plasmid or a plasmid expressing *ALKBH2* were treated with the indicated concentration of MMS following plating onto rich YPD medium. Colony number was assessed after incubation for 2 days at 30 °C. Fold rescue of cellular survival represents the ratio of the survival of cells expressing *ALKBH2* relative to the survival of cells expressing the empty plasmid. Survival represents the number of colonies as a percentage of the colonies obtained without MMS treatment. The individual and mean values from five to nine experiments were plotted. Error bars indicate 95 % confidence intervals. The p-values between WT and *csm2Δ* cells treated with 0.1 % MMS and 0.2 % MMS were calculated using an unpaired two-tailed Student's t-test and were $p \leq 0.01$ and $p \leq 0.001$, respectively. (**F**) S-phase *csm2Δ* cells expressing *ALKBH2* exhibit increased survival after acute MMS treatment. WT or *csm2Δ* cells were synchronized on G1 with alpha factor and either released from G1 arrest or maintained in G1 in the presence or absence of 0.1 % MMS. Cells were plated after 30 min of treatment and the colony number was assessed after incubation for 2 days at 30 °C. Survival is calculated as described in (**E**). The mean values from three experiments were plotted with standard deviations. The p-values between control (empty plasmid) and *ALKBH2* expressing cells were calculated using an unpaired two-tailed Student's t-test and were $p > 0.05$ (n.s.) and $p \leq 0.001$ for the G1- and S-phase cells, respectively.

The online version of this article includes the following figure supplement(s) for figure 3:

**Figure supplement 1.** Protein blot analysis of ALKBH2 and ALKBH2-CD expression in wild-type (WT) and *csm2Δ* cells.

**Figure supplement 1—source data 1.** Unprocessed images of the blots.

---

catalytic activity (***Figure 4A***; p < 0.0001). Consistent with the rescue of MMS sensitivity, the ALKBH2 rescue of MMS mutation rates is slightly, but not significantly, reduced in WT cells (***Figure 4A***; p = 0.0667). These spontaneous mutations are likely due to the TLS-mediated bypass of AP sites (***Ball et al., 2009***; ***Godin et al., 2016b***; ***Rosenbaum et al., 2019***; ***Shor et al., 2005***; ***Shrivastav et al., 2010***). To determine whether the decrease in mutations in MMS-treated, *ALKBH2*-expressing *csm2Δ* cells corresponds to the repair of 3meC, we sequenced the *CAN1* gene of canavanine-resistant clones. Can-R isolates from MMS-treated *csm2Δ* cells displayed frequent mutation of the *CAN1* gene with substitutions at A:T and C:G bases as well as single nucleotide deletions and complex events consisting of two mutations separated by less than 10 bp that are a signature of error-prone trans-lesion synthesis (***Northam et al., 2010***; ***Sakamoto et al., 2007***). Accordingly, *ALKBH2* expression in *csm2Δ* cells decreased the frequency of deletions, complex mutations, and substitutions at C:G base pairs (***Figure 4B***). These findings support a model in which the Shu complex prevents the cytotoxicity and mutagenicity of 3meC and in its absence, yeast is reliant on TLS to bypass these lesions, resulting in an increase of C:G substitutions, small deletions, and complex mutations.

We previously demonstrated that the Shu complex binds to double-flap DNA substrates and has a two-fold higher affinity for double-flap DNA containing an AP site analog (***Godin et al., 2016b***; ***Rosenbaum et al., 2019***). Therefore, we asked whether the DNA-binding subunits of the Shu complex, Csm2-Psy3, would similarly have a higher affinity for double-flap DNA-containing 3meC or 1meA, which also would result in A to T or A to G mutations. To address this, we examined the affinity of recombinant Csm2-Psy3 to a FAM-labeled double-flap substrate in the presence or absence of 3meC and 1meA by performing equilibrium titrations and measuring fluorescence anisotropy (***Figure 4C***). While Csm2-Psy3 binds to a double-flap substrate containing a 3meC or 1meA lesion on the ssDNA, it does so with lower affinity relative to unmodified substrate [3meC, $K_d = 252 \pm 21$; 1meA $K_d = 313 \pm 27$; unmodified $K_d = 180 \pm 29$ ***Figure 4C***]. These results suggest that while the Shu complex

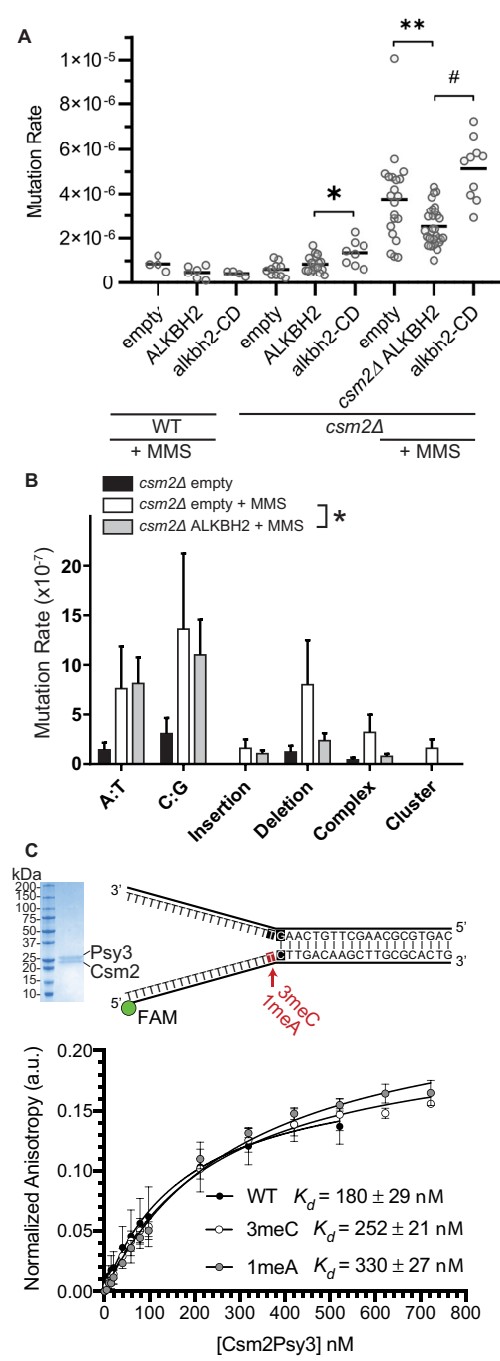

**Figure 4.** *ALKBH2* expression rescues the 3meC-induced mutagenesis observed in methyl methanesulfonate (MMS) exposed *csm2Δ* cells. (**A**) *csm2Δ* cells expressing *ALKBH2* exhibit reduced MMS-induced mutation rate. Spontaneous and MMS-induced mutation rates at the *CAN1* locus were measured in wild-type (WT) (MMS treated only) and *csm2Δ* cells transformed with an empty plasmid, a plasmid-expressing *ALKBH2* or *alkbh2-CD*. Each measurement represents a separate experiment (circle) and the median value (horizontal bar) of 4–22 experiments were plotted. The p-values were

*Figure 4 continued on next page*

*Figure 4 continued*

calculated using a Mann-Whitney ranked sum test and were as follows: p = 0.0667 for treated WT empty and WT ALKBH2; p = 0.1327 for untreated *csm2Δ* empty and *csm2Δ* ALKBH2; p = 0.0221 for untreated *csm2Δ* ALKBH2 and *csm2Δ* alkbh2-CD (*); p = 0.0092 for treated *csm2Δ* empty and *csm2Δ* ALKBH2 (**); and p < 0.0001 for treated *csm2Δ* ALKBH2 and *csm2Δ* alkbh2-CD (#). (**B**) Sequencing of the *CAN1* gene in canavanine-resistant colonies was used to calculate the frequency of MMS-induced substitutions, insertions, deletions, and complex mutations in *csm2Δ* cells transformed with either an empty vector or *ALKBH2* expression vector. ALKBH2 expression significantly alters the MMS-induced mutation spectra (* indicates p = 0.015 by chi-square analysis comparing the number of A:T substitutions, C:G substitutions, insertions, deletions, complex, and clustered mutations between MMS-treated *csm2Δ* cells containing an empty vector or *ALKBH2* expression vector). The spectrum contains a reduction in C:G substitutions, deletions, and complex mutations, while A:T substitutions and insertions are unchanged. (**C**) The Csm2-Psy3 protein binds to a double-flap DNA substrate containing unmodified, 3meC, and 1meA lesions. A Coomassie-stained SDS-PAGE gel of recombinant Csm2-Psy3, which run at 27.7 and 28 kDa, respectively. Equilibrium binding titrations were performed by titrating Csm2-Psy3 into the FAM-labeled double-flap substrate in unmodified ($K_d$ = 180 ± 29), 3meC containing ($K_d$ = 252 ± 21) and 1meA ($K_d$ = 313 ± 27) containing lesions at the indicated position, and anisotropy was measured. Experiments were performed with three protein preparations and standard deviations plotted. The data were fit to a quadratic equation (one-site binding model assumed) and dissociation constants ($K_d$) were calculated.

The online version of this article includes the following figure supplement(s) for figure 4:

**Source data 1.** Numerical data corresponding to the graph in panel A.

**Source data 2.** Unprocessed image of the gel in panel C.

**Source data 3.** Unprocessed anisotropy values.

binds double-flap substrates with 3meC or 1meA, Csm2-Psy3 preferentially recognizes a double-flap substrate containing an AP site mimetic.

## 3meC repair in yeast is channeled through error-free post-replicative repair

The Shu complex directly functions with the canonical Rad51 paralogs, Rad55-Rad57, and Rad52 to promote HR through Rad51 filament formation (*Gaines et al., 2015*; *Godin et al., 2013*). In the context of error-free post-replicative repair, the Shu complex functions downstream of

poly-ubiquitination of PCNA by Rad5-Ubc13-Mms2 (*Xu et al., 2013*). Therefore, we asked whether the rescue of the MMS-induced phenotypes by *ALKBH2* depends on the Shu complex function as a Rad51 mediator. To address this question, we utilized a Csm2 mutant, *csm2-F46A*, that cannot stimulate Rad51 filament formation due to the loss of its protein interaction with Rad55-Rad57 (*Gaines et al., 2015*; *Godin et al., 2013*). Suggesting that the Shu complex mediator function enables bypass of 3meC, we find that *ALKBH2* expression suppresses the MMS sensitivity of a *csm2-F46A* mutant to the same extent as a *csm2Δ* cell (*Figure 5A*). Next, we asked whether the rescue of MMS sensitivity by *ALKBH2* would be observed in other HR mutants. To accomplish this, we ectopically expressed *ALKBH2* in cells with deletions of *CSM2, RAD51, RAD52, RAD55,* and *UBC13* and performed serial dilutions upon increasing-MMS dosage (*Figure 5B*). Surprisingly, the MMS sensitivity of *rad51Δ*, *rad52Δ,* and *rad55Δ* is not rescued by *ALKBH2* expression to the same extent as the MMS sensitivity of *csm2Δ* cells. However, *rad51Δ* cells mutation rates are similarly reduced upon ALKBH2 expression (*Figure 5C*; compare *csm2Δ* ALKBH2-dependent rescue in *Figure 4A* to *rad51Δ* ALKBH2-dependent rescue in *Figure 5C*). In contrast to the Rad51 mediators, *ubc13Δ* MMS sensitivity is largely rescued by *ALKBH2* expression (*Figure 5B*). This striking result suggests that the Shu complex functions primarily within the Ubc13-initiated post-replicative repair pathway, while the canonical HR genes are also critical for the repair of DSBs induced by MMS through clustered lesions. Unlike Shu complex mutant cells, deletion of the canonical HR genes leaves cells vulnerable to toxic MMS-induced DSBs formed from lesions that are not subject to *ALKBH2* direct reversal.

To explore this idea further, we investigated the effect of *ALKBH2* in a *RAD55* phosphorylation mutant that is MMS sensitive while maintaining its DSB repair proficiency (*Herzberg et al., 2006*). In this *RAD55* mutant, three serine residues (2,8,14) are mutated to alanine residues (*rad55-S2,8,14A*). Since the *rad55-S2,8,14A* mutant cells largely phenocopy the defects observed in a Shu complex mutant (*Herzberg et al., 2006*), we asked whether Rad55 function in MMS-induced DNA damage may be uncoupled from its role in canonical DSB repair. Unlike *rad55Δ* cells, the *rad55-S2,8,14A* mutant MMS sensitivity is largely rescued by *ALKBH2* expression (*Figure 5D*). To further investigate the genetic relationship between the Shu complex and Rad55, we combined either *rad55Δ* or *rad55-S2,8,14A* with a *csm2Δ* mutant. As previously reported, *csm2Δ rad55Δ* double mutants exhibit the same MMS sensitivity as a *rad55Δ* mutant cell (*Godin et al., 2013*). In contrast, a *csm2Δ rad55-S2,8,14A* double mutant exhibits the same MMS sensitivity and mutation rates as a single *csm2Δ* cell (*Figure 5E and F*). This observation is surprising since the Shu complex is thought to function downstream of Rad55. However, it is consistent with the specificity of the Shu complex in directly recognizing and enabling tolerance of MMS-induced DNA lesions (*Rosenbaum et al., 2019*).

Rad55 directly interacts with Csm2-Psy3 (*Gaines et al., 2015*; *Godin et al., 2013*). While Rad55-S2,8,14A mutant maintains its protein interactions with Rad57, Rad51, and Rad52, its interaction with the Shu complex has yet to be determined (*Herzberg et al., 2006*). Therefore, the Shu complex may help recruit Rad55 to specific MMS-induced DNA lesions by interacting with phosphorylated Rad55 or at the interface where Rad55 is phosphorylated. To test this, we performed Y2H analysis of Rad55 or Rad55-S2,8,14A with Csm2 (*Figure 5G*). Interestingly, we observe a reduced interaction between Rad55-S2,8,14A with Csm2 (*Figure 5G*). Unfortunately, we were unable to confirm the Rad55-Csm2 interaction by co-immunoprecipitation, perhaps because the interaction is transient and is best observed by Y2H or in vitro pulldowns. These results suggest that Rad55 phosphorylation may stimulate its interaction with Csm2 or that Csm2 interacts with Rad55 in that region. In the context of MMS-induced DNA damage, our findings suggest that the Shu complex is likely contributing to the recognition of specific MMS-induced lesions and recruiting the HR machinery, such as Rad55-Rad57, to the lesion to facilitate their bypass and enable their repair following replication. Furthermore, other factors, such as a stalled replication fork, may also contribute to their recruitment.

## Discussion

Among the many alkylation-induced DNA lesions, 3meC is notable for its cytotoxic and mutagenic impact during DNA replication (*Nieminuszczy et al., 2009*; *Shrivastav et al., 2010*; *Sikora et al., 2010*). In eukaryotes, the adduct is formed endogenously from *S*-adenosyl methionine (SAM) reactivity with DNA and from the enzymatic activity of DNA methyltransferases, and exogenously from alkylating agents such as nitrosamines, which are present in the tobacco smoke, temozolomide, or MMS (*Chatterjee and Walker, 2017*; *Dango et al., 2011*; *Pataillot-Meakin et al., 2016*; *Rošić et al.,*

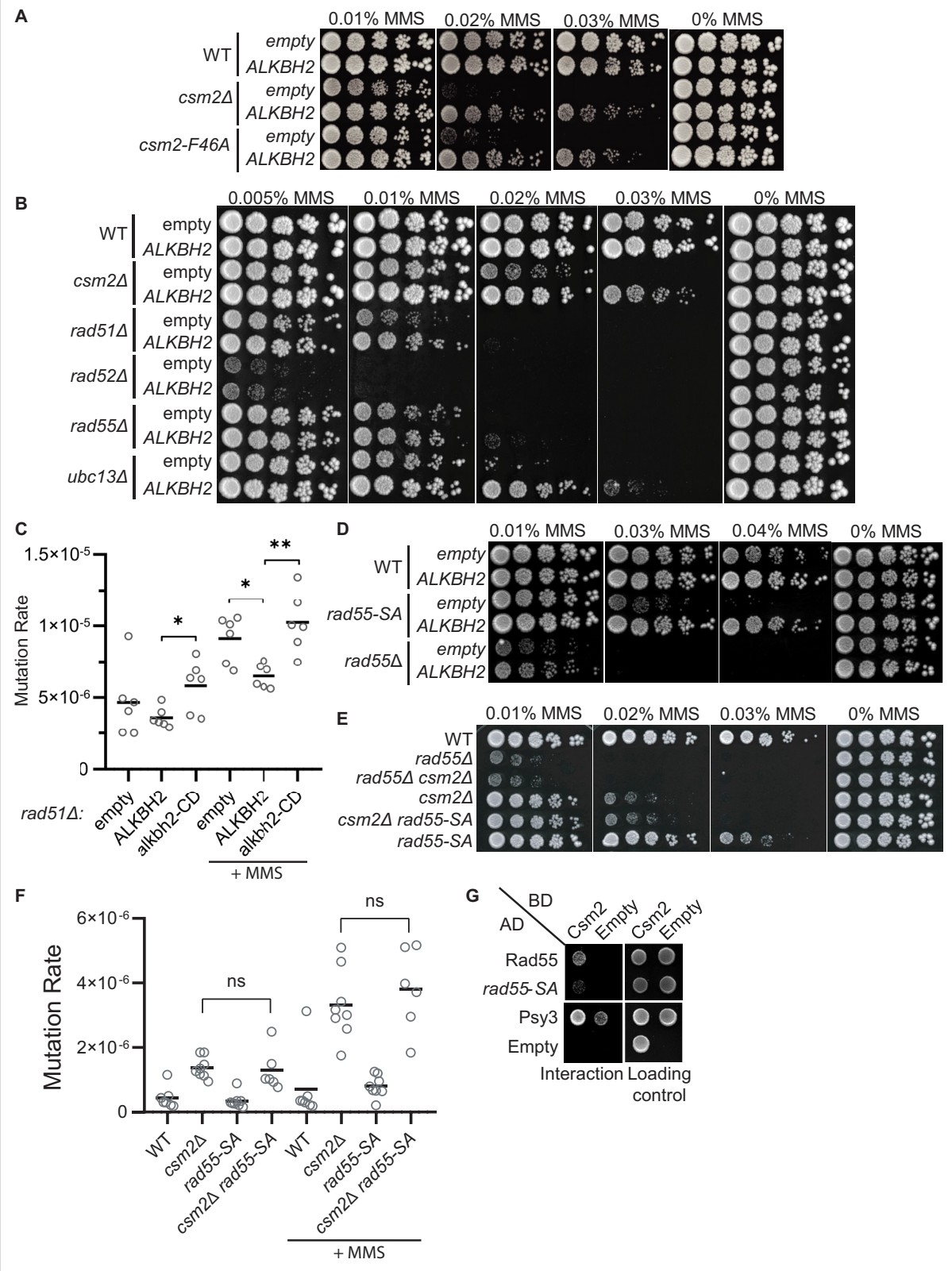

**Figure 5.** 3meC are bypassed by the error-free post-replication repair (PRR) pathway. (**A**) *ALKBH2* rescues the methyl methanesulfonate (MMS) sensitivity of a *csm2-F46A* mutant, which is deficient for its Rad51 mediator activity. Fivefold serial dilutions of wild-type (WT), *csm2Δ*, or *csm2-F46A* cells transformed with an empty plasmid or a plasmid-expressing *ALKBH2*, were platted onto rich YPD medium or YPD medium containing the indicated MMS concentration and incubated for 3 days at 30 °C prior to being photographed. (**B**) Unlike PRR mutant *UBC13*, expression of *ALKBH2* mildly rescues

*Figure 5 continued on next page*

*Figure 5 continued*

the MMS sensitivity of homologous recombination (HR) factors, *RAD51, RAD52,* and *RAD55*. Fivefold serial dilution of WT, *csm2Δ, rad51Δ, rad52Δ, rad55Δ,* or *ubc13Δ* transformed with an empty plasmid or a plasmid-expressing *ALKBH2* were fivefold serially diluted, plated, and analyzed as described in (**A**). (**C**) *rad51Δ* cells expressing *ALKBH2* exhibit reduced MMS-induced mutation rate. Spontaneous and MMS-induced mutation rates at the *CAN1* locus were measured in rad51Δ cells transformed with an empty plasmid, a plasmid-expressing *ALKBH2* or *alkbh2-CD*. Each measurement represents a separate experiment (circle) and the median value (horizontal bar) of six experiments were plotted. The p-values were calculated using a Mann-Whitney ranked sum test and were as follows: p = 0.5887 for untreated empty and ALKBH2; p = 0.026 untreated ALKBH2 and alkbh2-CD (*), p = 0.026 for MMS-treated *rad51Δ* empty and ALKBH2 (*); p = 0.0043 for MMS-treated *rad51Δ* ALKBH2 and alkbh2-CD (**). (**D**) *rad55-S2,8,14A* (*rad55-SA*) cells expressing *ALKBH2* exhibit decreased MMS sensitivity. WT, *rad55-S2,8,14A,* or *rad55Δ* cells transformed with an empty plasmid or a plasmid-expressing *ALKBH2* were fivefold serially diluted, plated, and analyzed as described in (**A**). (**E**) *csm2Δ* is epistatic to *rad55-S2,8,14A* (*rad55-SA*) for MMS damage. Cells with the indicated genotypes were fivefold serially diluted and plated as described in (**A**), and incubated for 2 days at 30 °C prior to being photographed. (**F**) The mutation rate of a *csm2Δ rad55-S2,8,14A* (*csm2Δ rad55-SA*) double mutant is the same as a *csm2Δ* cell. Spontaneous and MMS-induced mutation rates at the *CAN1* locus were measured in WT, *csm2Δ, rad55-SA,* and *csm2Δ rad55-SA* cells. Each measurement represents a separate experiment (circle) and the median value (horizontal bar) of six to eight experiments were plotted. The p-values were calculated using a Mann-Whitney ranked sum test and were p = 0.3238 and p = 0.3965 for *csm2Δ* and *csm2Δ rad55-SA* untreated or MMS-treated, respectively (not significant, ns). (**G**) rad55-S2,8,14A (*rad55-SA*) exhibit an impaired yeast-2-hybrid (Y2H) interaction with Csm2. Y2H analysis of pGAD-*RAD55*, *rad55-S2,8,14A*, *PSY3*, or pGAD-C1 (Empty) with pGBD-*RAD57*, *CSM2*, pGBD-C1 (Empty). A Y2H interaction is indicated by plating equal cell numbers on SC medium lacking histidine, tryptophan, and leucine. Equal cell loading is determined by plating on synthetic complete (SC) medium lacking tryptophan and leucine used to select for the pGAD (AD) and pGBD (BD) plasmids.

The online version of this article includes the following figure supplement(s) for figure 5:

**Source data 1.** Numerical data corresponding to the graph in panel C.

**Source data 2.** Numerical data corresponding to the graph in panel F.

*2018*). 3meC occurs primarily in ssDNA and can stall the replicative polymerases. Unlike bacteria and higher eukaryotes, yeast does not encode for an enzyme capable of directly repairing 3meC from ssDNA (*Admiraal et al., 2019*; *Sedgwick et al., 2007*). Therefore, yeast relies on bypass mechanisms to complete DNA replication. Here, we utilized the budding yeast model to demonstrate that the Shu complex facilitates an HR-mediated error-free bypass of alkylation damage, including 3meC and AP sites, to prevent mutagenesis and toxicity.

The Shu complex is primarily involved in the tolerance of replicative base-template damage, being dispensable for DSB repair (*Ball et al., 2009*; *Godin et al., 2016b*). This makes the Shu complex ideal to dissect the role of HR during bypass of specific base lesions. The major phenotypes observed in MMS-exposed Shu complex disrupted cells are decreased cell survival and elevated mutation frequency (*Ball et al., 2009*; *Godin et al., 2016c*; *Shor et al., 2005*). We observe that *ALKBH2* expression specifically alleviates the MMS-induced phenotypes of Shu complex disrupted cells (*Figures 3 and 4*). This *ALKBH2* rescue is partial, which is consistent with the Shu complex role in tolerance of another MMS-induced lesion, an AP site (*Godin et al., 2016a*; *Rosenbaum et al., 2019*). We also note that in some cases alkbh2-CD-expressing cells exhibit more mutations than the empty vector (*Figures 4A and 5C*). These results suggest that perhaps alkbh2-CD may interfere with the processing of 3meC. Although it is not possible to specifically induce 3meC, it is possible to regulate the occurrence of its template, by controlling the amount of ssDNA in MMS-exposed cells. It would be interesting to observe how the rescue by expression of *ALKBH2* correlates with the amount of ssDNA. Consistently, our results show that *ALKBH2* is only able to rescue the MMS sensitivity of Shu complex mutant cells that are progressing through S-phase and therefore exhibit ssDNA intermediates and when an HR-dependent error tolerance occur (*Figure 3F*).

When a replication fork stalls, the replicative helicase and polymerase can uncouple, and ssDNA can form behind the fork on the lagging strand. To analyze a fork-like substrate in vitro, we previously analyzed different types of forks including double-flap, 5' flap, 3' flap, and static flaps, and observe similar binding affinities (*Rosenbaum et al., 2019*). A double-flap substrate could parallel the uncoupling of the helicase and polymerase. We previously showed that the Shu complex bind to a double-flap substrate containing an AP site mimetic on the ssDNA lagging strand (*Rosenbaum et al., 2019*). While we did not observe increased binding affinity for a double-flap with 3meC and 1meA with Csm2-Psy3 (*Figure 4C*), other factors may contribute to binding such as the other Shu complex members or the Rad51 paralogs, Rad55-Rad57. In support of this model, a connection between the human RAD51 paralog, RAD51C, and ALKBH3 was recently described where RAD51C was important

for the recruitment of ALKBH3 to DNA with 3meC (*Mohan et al., 2019*).

The error-free PRR pathway requires polyubiquitination of PCNA by Mms2-Ubc13-Rad5 and the activity of the core HR machinery (*Branzei and Szakal, 2017*; *Xu et al., 2015*). Interestingly, the deletions of other HR factors, such as *RAD51*, are only partially rescued by *ALKBH2* expression (*Figure 5B*). This can be explained by their critical role in DSB repair, an activity for which the Shu complex is largely dispensable. Our results are consistent with 3meC being bypassed by the HR branch of the PRR pathway (*Figure 6*) and the notion that the Shu complex is an HR factor specialized in replication-associated damage.

It is important to note that in many ways, the mutation pattern of MMS-treated *csm2Δ* cells is an augmented version of MMS-induced mutations in WT cells. This similarity is likely explained by the Shu complex facilitating error-free bypass of multiple mutagenic alkylation lesions, which we highlight here prominently as 3meA and 3meC lesions. TLS can bypass 3meA directly and cause elevated A-> T substitutions (*Shrivastav et al., 2010*) seen in both MMS-treated WT and *csm2Δ* cells. However, previous work from our group has provided genetic, in vivo, and in vitro evidence of the Shu complex role in the error-free bypass of AP sites specifically (*Godin et al., 2016b*, *Rosenbaum et al., 2019*), suggesting some of the Shu complex's protection against mutations at A:T base pairs may be due to bypass of AP sites derived from spontaneous or glycolytic removal of the 3meA base.

Some MMS-induced A:T mutations in *csm2Δ* cells may be due to the failure of the Shu complex and the error-free PRR pathway to bypass 1meA. AlkB proteins are able to repair 1meA, which like 3meC primarily occurs at ssDNA (*Fedeles et al., 2015*) and is a toxic and mutagenic adduct (*Shrivastav et al., 2010*). Hence, we cannot rule out that ALKBH2 repair of 1meA may also contribute to the rescue of MMS-induced phenotypes that we observe, and that 1meA is a source of some MMS-induced mutations in both WT and *csm2Δ* cells. However, the substitution patterns and transcriptional asymmetry observed for unselected A mutations in MMS-treated, whole genome sequenced *csm2Δ* yeast as well as the similarity in the *CAN1* mutation spectra of *mag1Δ* and *mag1Δ csm2Δ* cells are all consistent with 3meA lesions being the primarily mutagenic lesion in A:T bases that are suppressed by the Shu complex. Moreover, our whole genome sequence analysis indicates a roughly equal amount of MMS-induced

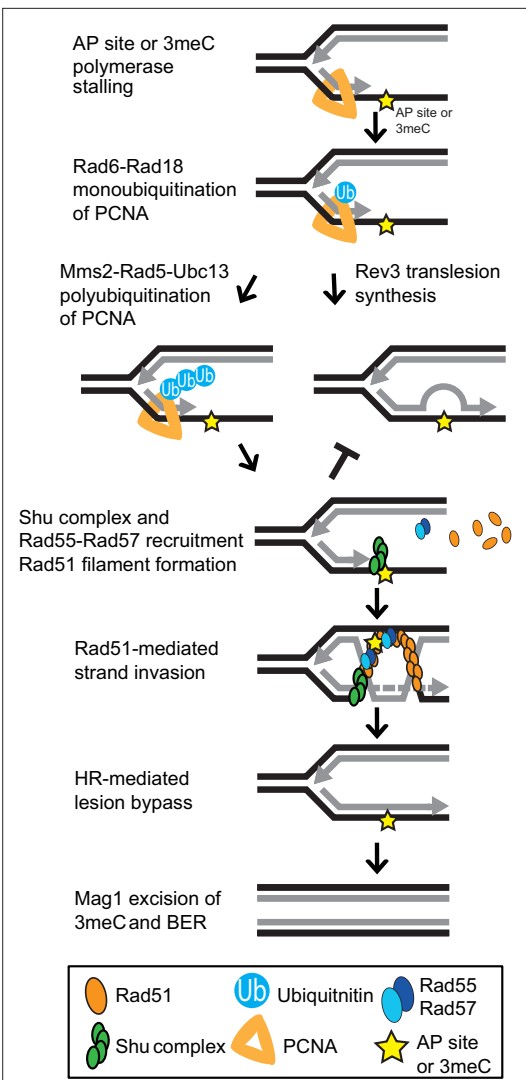

**Figure 6.** Model of Shu complex-mediated error-free bypass of abasic (AP) sites and 3meC. MMS-induced AP sites and 3meC (yellow star) arising at DNA replication intermediates at ssDNA can stall the replicative polymerase. Replication fork stalling leads to PCNA (orange triangle) K63-linked polyubiquitination of lysine 164 (K164) by the sequential activities of the Rad6-Rad18 and Mms2-Rad5-Ubc18 complexes. When an AP site or 3meC forms, the Shu complex (green ovals) promotes Rad55-Rad57 (blue ovals) recruitment and Rad51 filament formation (orange ovals). Thus, enabling Rad51-mediated HR with the newly synthesized sister chromatid. Importantly, the Shu complex activity prevents mutagenesis from TLS-mediated error-prone bypass of 3meC. After DNA synthesis using the undamaged sister chromatid as a template, the HR intermediates are resolved. The error-free bypass of 3meC enables S-phase completion in a timely manner. Finally, after replication is completed, 3meC are likely recognized and excised by the Mag1 glycosylase, which initiates the BER-mediated repair.

*Figure 6 continued on next page*

*Figure 6 continued*

The online version of this article includes the following figure supplement(s) for figure 6:

**Figure supplement 1.** *TPA1* does not genetically interact with *CSM2* or *MAG1* for methyl methanesulfonate (MMS) damage.

mutations at A:T and G:C base pairs. Since 3meC is often observed as the main lesion contributing to mutagenesis from MMS in ssDNA (*Saini et al., 2020*; *Yang et al., 2010*) (i.e. 3meC produces more mutations than 1meA), then one would expect MMS-induced G:C mutations to predominate if 1meA was the primarily mutagenic lesion at A:T lesions that the Shu complex helps to bypass. Thus, the similarity in the A:T substitution patterns induced by MMS in WT and *csm2Δ* yeast is likely due to the majority of mutations stemming from 3meA lesions in both cases.

Whether the similarity in the MMS-induced G:C substitution pattern in WT and *csm2Δ* cells is likewise due to the mutations in both cases stemming from the same lesions is less clear. While the observed substitution pattern for MMS-induced G:C substitutions is generally inconsistent with a mutagenic mechanism, 7meG lesions are converted to AP sites prior to substitution event (*Figure 1C*) and 7meG lesions themselves are frequently thought to be non-mutagenic (*Shrivastav et al., 2010*), recent determination of MMS-induced mutation spectra in ssDNA indicates that some base substitutions in WT yeast are caused by lesions on G bases (*Saini et al., 2020*). Based on the expected lesion abundance (*Drabløs et al., 2004*), these mutations would likely be caused by 7meG directly or due to the combined effects of several of its spontaneously formed chemical derivatives. This mutation spectrum is complementary to previously established spectra for 3meC-induced mutations in yeast (*Roberts et al., 2012*; *Saini et al., 2020*; *Yang et al., 2010*), making it difficult to assess whether G:C substitutions in MMS-treated WT yeast or *csm2Δ* cells are due to 7meG lesions or 3meC lesions formed in transient ssDNA based on substitution patterns alone. In the case of MMS-induced mutations in *csm2Δ* cells, the additional transcriptional and replicative asymmetries are suggestive of the Shu complex protecting against mutations induced by 3meC (i.e. more frequent mutation of C in strands expected to be more often transiently single-stranded). The ability to observe this strand bias requires the lesions causing the bias must constitute a relatively large portion of the G:C substitutions; and the additional suppression of MMS-induced toxicity in *csm2Δ* cells by ALKBH2 expression confirms that at least some of the mutations are 3meC-derived. However, evidence of strand bias does not exclude the possibility that 7meG also induces error-free lesion bypass mediated through the Shu complex and contributes to the MMS-induced mutation spectra observed in WT and *csm2Δ* yeast.

A previous study claimed that yeast *TPA1* is an AlkB homolog (*Shivange et al., 2014*). However, in our hands, and consistent with a recent report (*Admiraal et al., 2019*), *tpa1Δ* cells show no increased sensitivity to MMS. Furthermore, unlike other genes that are involved in repairing MMS-induced lesions, we find that *TPA1* does not exhibit a synthetic sick phenotype with Shu complex mutants upon MMS exposure (*Figure 6—figure supplement 1*). Recently, the Mag1 glycosylase was shown to excise 3meC in dsDNA, therefore, initiating their repair through BER (*Admiraal et al., 2019*). However, Mag1 cannot excise 3meCs from ssDNA (*Admiraal et al., 2019*). Whole genome sequencing of MMS-treated *mag1Δ* yeast also contains primarily an elevation of substitutions at A:T base pairs indicating that Mag1 activity in yeast contributes little to protecting against 3meC-induced mutations. This is likely due to the restriction of these lesions to ssDNA and highlights the need for a mechanism to bypass this lesion during DNA replication (*Figure 6*). Mag1 likely plays a significant role in the removal of 3meC from dsDNA after its Shu complex-mediated bypass allows replication to be completed (*Figure 6*).

Previous work from our group and others shows that the Shu complex role is functionally conserved in humans and mice (*Abreu et al., 2018*; *Martino and Bernstein, 2016*; *Martino et al., 2019*). Additionally, human cells containing Shu complex deletions are sensitive to MMS and DNA alkylating agents. Therefore, future studies will address whether the human Shu complex functions as a backup of the ALKBH enzymes in tolerating mutagenesis and toxicity from 3meC. This is of particular importance since both *ALKBH2* and *ALKBH3* have been proposed as tumor suppressors, being silenced in various tumors, including gastric and breast cancer (*Fedeles et al., 2015*; *Gao et al., 2011*; *Knijnenburg et al., 2018*; *Stefansson et al., 2017*). On the other hand, *ALKBH3* is often overexpressed in different cancers and inhibition of *ALKBH2* and *ALKBH3* can sensitize cancer cells to alkylating chemotherapy (*Choi et al., 2011*; *Koike et al., 2012*; *Tasaki et al., 2011*; *Wang et al., 2015*; *Wu*

*et al., 2011*). Moreover, the upregulation of *ALKBH2* and *ALKBH3* mediates the resistance to chemo-therapeutic agents such as temozolomide and *ALKBH3* loss leads to endogenous 3meC accumulation in tumors cell lines (*Cetica et al., 2009*; *Dango et al., 2011*; *Johannessen et al., 2013*). A role for the Shu complex in promoting tolerance of 3meC could provide a new avenue for therapeutic approaches to target these tumors.

## Materials and methods
### Yeast strains, plasmids, and oligos
The strains utilized are listed in *Supplementary file 1a* whereas all oligonucleotides used are listed in *Supplementary file 1b*. The Y2H strains PJ69-4A and PJ69-4α were used as described (*Godin et al., 2013*; *James et al., 1996*). All strains are isogenic with W303 RAD5+ W1588-4 C (*Thomas and Rothstein, 1989*) and W5059-1B (*Zhao et al., 1998*). KBY-1088-3 C (rad55-S2,8,14A) was generated by transformation of a cassette containing the 50 bp homology upstream of the *RAD55* start codon and the rad55-S2,8,14A ORF fused to a kanMX6 resistance cassette and the 50 bp homology downstream of the *RAD55* stop codon. This fused cassette was obtained using Gibson Assembly Master Mix (NEB) following the manufacturer's instructions and the primers used to generate the assembly fragments were designed using NEBuilder Assembly Tool (https://nebuilder.neb.com). The rad55-S2,8,14A gene fragment was commercially synthesized whereas the kanMX6 cassette was amplified from the pFA6a-kanMX6 plasmid (*Longtine et al., 1998*). Prior to transformation, the fused product was PCR-amplified using primers with 50 nt of homology to the flanking regions of *RAD55* as described (*Longtine et al., 1998*). All yeast transformations were performed as described (*Sherman et al., 1986*). The integration of the rad55-S2,8,14A ORF was verified by PR amplification followed by sequencing using the KanHisNat and Rad55. Check oligos as described in *Longtine et al., 1998*. pGAD-rad55-S2,8,14A was generated using Gibson Assembly Master Mix (NEB) following the manufacturer's instructions and the primers used to generate the assembly fragments were designed using NEBuilder Assembly Tool (https://nebuilder.neb.com). ALKBH2 and ALKBH3 were cloned in the pAG416GPD-ccdB vector (Plasmid #14148 Addgene). pAG416GPD-ccdB-ALKBH2-CD was generated by site-directed muta-genesis of the pAG416GPD-ccdB-ALKBH2 plasmid as described (*Zheng et al., 2004*) with minor adaptations according to the manufacturer's recommendations for PCR using Phusion High-Fidelity PCR Master Mix with HF Buffer (Thermo). All knock-outs were generated using the S1 and S2 primers and knock-out cassettes as described in *Longtine et al., 1998*. All plasmids and strains were verified by DNA sequencing of the cloned open reading frame.

### Chronic MMS exposure and DNA whole genome sequencing
Individual colonies of WT or *csm2Δ* cells were grown overnight at 30 °C. The cultures were then pinned onto YPD medium containing 0.008 % MMS using a pinning robot from S&P Robotics. After a 2 -day incubation at 30 °C, the plates were replica-plated onto YPD plates containing 0.008 % MMS using a robotic pinner and then replated onto fresh YPD medium containing 0.008 % MMS a total of 10 times. The MMS-exposed yeasts were separated into single colonies (96 per strain). These colonies were inoculated in YPD cultures and grown overnight at 30 °C. Genomic DNA was extracted from each culture by resuspending the yeast pellets in lysis buffer (20 mM Tris-HCl, 200 mM LiAc, 1.5 % SDS, pH 7.4), incubating the yeast for 15 min at 70 °C, incubating on ice for 10 min, adding an equal volume of 4 M NaCl, and centrifuging the samples at maximum speed for 30 min at 4 °C. The supernatant was then added to an equal volume of phenol:chloroform:isoamyl alcohol (PCI). An equal volume of isopropanol was added to the aqueous phase of the PCI extraction. The DNA pellet was washed twice with 70 % ethanol prior to resuspending in 10 mM Tris 8 buffer. The DNA was subjected to Illumina whole genome sequencing; 1 μg of genomic DNA per sample was sheared and used to generate libraries for sequencing with a KAPA DNA HyperPrep kit. Multiplexed libraries were sequenced with 96 samples on a single lane of an Illumina Hiseq4000. Illumina reads were aligned to the Saccer3 S288C reference genome with CLC Genomics Workbench version 7.5. CLC Genomics Workbench was also used to identify mutations from these alignments using methods similar to those previously described (*Sakofsky et al., 2014*). Briefly, mutations were identified as differences from the reference genome that were greater than nine reads covering the site, and for which between 45% and 55% of the reads supported the mutation. Additionally, any mutations occurring in multiple samples were

removed from analyses as likely polymorphisms or alignment artifacts. The complete list of mutations is provided in *Supplementary file 1c*. Raw sequencing reads in fastq format have been submitted to the NCBI short read archive under BioProject accession number PRJNA694993.

## Growth assays

Individual colonies of the indicated strains were transformed with an empty plasmid or a plasmid-expressing ALKBH2 as indicated. The cultures were grown in 3 ml YPD or synthetic complete (SC)-URA medium overnight at 30 °C. Fivefold serial dilutions were performed (*Godin et al., 2016b*) except that 5 µl of culture at 0.2 $OD_{600}$ were fivefold serially diluted onto YPD medium or YPD medium containing the indicated MMS concentration. UV treatment was performed using Stratagene Stratalinker 2400 UV Crosslinker. The plates were imaged after 48 or 72 hr of incubation at 30 °C and the brightness and contrast were globally adjusted using Photoshop (Adobe Systems Incorporated).

## Protein blotting

Parental and *csm2Δ* cells were transformed with an empty plasmid or plasmid expressing either ALKBH2 or ALKBH2-CD. The cultures were grown in SC-URA medium overnight at 30 °C. Subsequently, equal cell numbers (1 ml 0.75 $OD_{600}$) from each culture were pelleted, the supernatant was removed, and washed once with ddH$_2$O, and pelleted again. Protein was extracted from whole cell lysates by TCA preparation as described in 51 µl of loading buffer (*Knop et al., 1999*). The 13 µl of protein was run on a 10 % SDS-PAGE gel and transferred to a PDVF membrane by semidry transfer (Bio-Rad) at 13 V for 90 min. ALKBH2 or Kar2 was Western blotted using αALKBH2 antibody (AbCam [ab154859]; 1:1000 with secondary antibody anti-rabbit [Jackson ImmunoResearch Laboratories 1:10,000]) or αKar2 antibody (Santa Cruz [sc-33630]; 1:5000 with secondary antibody anti-rabbit [Jackson ImmunoResearch Laboratories 1:10,000] as a loading control).

## Survival assays

Individual colonies of WT or *csm2Δ* cells were transformed with an empty plasmid or a plasmid-expressing *ALKBH2* and grown in 3 ml SC-URA medium at 30 °C overnight. The cultures were diluted to 0.2 $OD_{600}$ in 50 ml SC-URA medium and grown for 3–4 hr at 30 °C. The cultures were all diluted to 0.2 $OD_{600}$ in YPD or YPD containing 0.05%, 0.1 %, or 0.2 % MMS and incubated for 30 min at 30 °C. After the treatment, the cultures were washed twice with YPD and resuspended to 0.2 $OD_{600}$ in YPD. The cultures were diluted 1/10,000 (untreated and 0.05 % MMS) or 1/1000 (0.1% and 0.2% MMS) and 150 µl were plated in YPD medium plates in duplicate. The plates were incubated at 30 °C for 2 days before imaging. The colonies were counted using OpenCFU (*Geissmann, 2013*). Data from five to nine colonies from at least three independent transformants was used.

## Canavanine mutagenesis assay

Individual colonies of the indicated strains were transformed with an empty plasmid or a plasmid-expressing *ALKBH2* and grown in 3 ml SC-URA medium or SC-URA medium containing 0.00033 % MMS for 20 hr at 30 °C. The cultures were diluted to 3.0 $OD_{600}$. 150 µl were plated on SC-ARG+ CAN (0.006 % canavanine) medium in duplicate or 150 µl of a 1:10,000 dilution were plated on SC medium in duplicate. The plates were then incubated for 48 hr at 30 °C before imaging. The colonies were counted using OpenCFU (*Geissmann, 2013*) to measure total cell number (SC) or forward mutation rates (SC-ARG+ CAN). The mutation frequency was obtained by dividing colony number in SC-ARG+ CAN by the number obtained in the SC plates times the dilution factor. Mutation frequencies were converted to mutation rates using methods described (*Drake, 1991*). Data from 10 to 20 colonies, from at least three independent transformants, was used. Differences in mutation rates were evaluated using a Mann-Whitney rank sum test comparing the independent rate measurements. Mutations occurring within the *CAN1* gene were identified by selecting independent canavanine resistant colonies, isolating genomic DNA from these samples, and PCR amplifying the *CAN1* gene from each isolate with primers containing unique barcodes (*Supplementary file 1d*). These PCR products were pooled and sequenced on a PacBio Sequel sequencer as in *Hoopes et al., 2017* or by traditional Sanger sequencing. PacBio reads were separated by barcodes and both PacBio and Sanger reads aligned to the *CAN1* gene sequence using Geneious software. Geneious was also used to identify mutations among these alignments as differences from the reference *CAN1* sequence that are

supported by two or more independent reads that occur in greater than 30 % of all reads for each sample; 947 out of 997 mutations identified were supported by greater than 50 % of the reads for the sample. All mutations identified by Sanger and PacBio sequencing are provided in *Supplementary file 1e*.

## Yeast-two-hybrid assays

The yeast-two-hybrid experiments using the indicated pGAD and pGBD plasmids were performed (*Godin et al., 2013*) except that both pGAD and pGBD plasmids were transformed into PJ69-4A (*James et al., 1996*). A yeast-two-hybrid interaction is indicated by growth on synthetic complete medium lacking histidine, tryptophan, and leucine whereas equal cell loading was observed by plating the cells on SC medium lacking tryptophan and leucine to select for the pGAD (leucine) or pGBD (tryptophan) plasmids.

## Csm2-Psy3 purification

The Csm2-Psy3 heterodimer was cloned into the dual expression plasmid pRSFDuet (EMD Millipore) which encode a Strep-tagged Csm2 and FLAG-tagged Psy3. All affinity tags were fused to the N-terminus of the protein. This plasmid was transformed into *Escherichia coli* (Rosetta P.LysS) and grown at 37 °C until 0.6 $OD_{600}$ and recombinant protein expression was induced by the addition of 0.5 mM isopropyl beta thiogalactoside at 18 °C overnight for 16–18 hr. Cells were harvested by centrifugation and pellets were frozen at –80 °C. Approximately 38 g of cell pellet was lysed in 80 ml of lysis buffer containing 25 mM Tris (pH 7.4), 300 mM KCl, 10 % glycerol, 5 mM β-mercaptoethanol supplemented with protease inhibitors (Roche), 1 mM PMSF, 1 mM $ZnCl_2$, and 0.01 % IGEPAL CA-630. Cells were lysed using an emulsiflex and centrifuged at 15,000× *g* for 1 hr at 4 °C. Lysed supernatant was incubated with 5 ml of ANTI-FLAG M2 affinity resin (Sigma) for 1 hr and then washed in a solution consisting of 25 mM Tris (pH 7.4), 150 mM KCl, 10 % glycerol, and 0.01 % IGEPAL CA-630. To elute bound Csm2-Psy3, the wash buffer was supplemented with 0.1 mg/ml 3X-FLAG peptide (Sigma). Elutions were then loaded onto a HiTrap Heparin HP (GE Healthcare) affinity chromatography. The complex was loaded onto the column and washed for 200 ml with (25 mM Tris (pH 7.4), 150 mM KCl, 1 mM DTT, 10 % glycerol, and 0.01 % IGEPAL CA-630). The complex was eluted with a gradient elution from 0% to 100% (25 mM Tris [pH 7.4], 150 mM KCl, 1 mM DTT, 10 % glycerol, and 0.01 % IGEPAL CA-630) over 20 ml. The Csm2-Psy3 protein is typically eluted around 400–600 mM NaCl. Then, Csm2-Psy3 heparin elutions were subsequently purified by size exclusion chromatography using a Sephacryl S200 column (GE Healthcare) in buffer (25 mM Tris [pH 7.4], 1 M KCl, 1 mM DTT, 10 % glycerol, and 0.01 % IGEPAL CA-630) eluting as a single peak and visualized as heterodimers by SDS-PAGE electrophoresis. Csm2-Psy3 protein concentration was determined by BCA assay as previously described (*Rosenbaum et al., 2019*).

## Substrates for fluorescence anisotropy

| | |
|---|---|
| WT | 5'-CAGTGCGCAAGCTTGTCAAGTTTTTTTTTTTTTTTTTTTTT-3' |
| WTT-5'FAM | 5'/56-FAM/TTTTTTTTTTTTTTTTTTTTCCTTGACAAGCTTGCGCACTG-3' |
| 3meC-5'FAM | 5'/56-FAM/TTTTTTTTTTTTTTTTTTTT/i3Me-dC/CTTGACAAGCTTGCGCACTG-3' |
| 1meA-5'FAM | 5'/56-FAM/TTTTTTTTTTTTTTTTTTTTT/i1Me-dA/CTTGACAAGCTTGCGCACTG-3' |

## Equilibrium fluorescence anisotropy assays

Anisotropy experiments were performed using a FluoroMax-3 spectrofluorometer (HORIBA Scientific) and a Cary Eclipse Spectrophotometer. All substrates used for anisotropy contained a 5'-FAM moiety incorporated at the end of the single-stranded end of the fork (*Figure 4C*). Substrates either contained a thymidine (T), 3meC, or 1meA at the double-flap junction in the ssDNA. Anisotropy measurements were recorded in a 150 µl cuvette containing 20 mM Tris pH 8.0 and 5 nM of FAM-labeled double-flap substrate as a premixed sample of purified Csm2-Psy3 protein and substrate (5 nM WT + 1.6 µM Csm2-Psy3 dimer in 50 nM NaCl) was titrated into the cuvette. Fluorescence anisotropy measurements were recorded using the integrated polarizer and excitation and emission wavelengths of 495 and 520 nm, respectively, with slit widths of 10 nm. Titrations were carried out

at 25 °C and were carried out until anisotropy became unchanged. All experiments were performed in triplicate with three protein preparations. Dissociation constants ($K_d$) were calculated by fitting our data to a quadratic equation [$Y = M*((x + D + K_d) - \text{sqrt}(((x + D + K_d)^2) - (4*D*x)))/(2*D)$] assuming a one-site binding model. Data were fit with PRISM7 software. Unprocessed raw anisotropy values are in the source data file.

## Acknowledgements

This study was supported by the National Institutes of Health grant (ES030335 to KAB; CA218112 to SAR) and the American Cancer Society (129182-RSG-16-043-01-DMC to KAB and 133947-PF-19-132-01-DMC to SRH). This work was supported by Hillman Fellows for Innovative Cancer Research Program.

## Additional information

### Funding

| Funder | Grant reference number | Author |
|---|---|---|
| National Institutes of Health | ES030335 | Kara A Bernstein |
| National Institutes of Health | CA218112 | Steven A Roberts |
| American Cancer Society | 129182-RSG-16-043-01-DMC | Kara A Bernstein |
| American Cancer Society | 133947-PF-19-132-01-DMC | Sarah R Hengel |

The funders had no role in study design, data collection and interpretation, or the decision to submit the work for publication.

### Author contributions

Braulio Bonilla, Conceptualization, Data curation, Formal analysis, Investigation, Methodology, Writing – original draft, Writing – review and editing; Alexander J Brown, Sarah R Hengel, Kyle S Rapchak, Debra Mitchell, Catherine A Pressimone, Adeola A Fagunloye, Thong T Luong, Reagan A Russell, Rudri K Vyas, Tony M Mertz, Ewa P Malc, Piotr A Mieczkowski, Investigation; Hani S Zaher, Conceptualization, Resources, Writing – review and editing; Nima Mosammaparast, Conceptualization, Methodology, Resources, Writing – review and editing; Steven A Roberts, Conceptualization, Formal analysis, Funding acquisition, Investigation, Methodology, Project administration, Resources, Supervision, Writing – original draft, Writing – review and editing; Kara A Bernstein, Conceptualization, Formal analysis, Funding acquisition, Methodology, Project administration, Writing – original draft, Writing – review and editing

### Author ORCIDs

Braulio Bonilla http://orcid.org/0000-0001-7468-9232
Adeola A Fagunloye http://orcid.org/0000-0003-2383-9469
Hani S Zaher http://orcid.org/0000-0002-7424-3617
Steven A Roberts http://orcid.org/0000-0002-3628-5808
Kara A Bernstein http://orcid.org/0000-0003-2247-6459

### Decision letter and Author response

Decision letter https://doi.org/10.7554/eLife.68080.sa1
Author response https://doi.org/10.7554/eLife.68080.sa2

## Additional files

### Supplementary files

• Supplementary file 1. (a): List of strains used in this study (b): PCR oligonucleotides used in this

study (c): Unique mutations in diploid wild-type (WT) and csm2Δ/csm2Δ methyl methanesulfonate (MMS) exposed yeast (d): Primers used for CAN1 sequencing (e): CAN1 sequencing from untreated and methyl methanesulfonate (MMS)-treated yeast.

- Transparent reporting form

### Data availability

All unique mutations identified by DNA sequencing are reported in Supplementary File 1c and all sequencing reads are reported in Supplementary file 1e. Raw sequencing reads in fastq format have been submitted to the NCBI short read archive under BioProject accession number PRJNA694993.

The following dataset was generated:

| Author(s) | Year | Dataset title | Dataset URL | Database and Identifier |
|---|---|---|---|---|
| Bonilla B, Brown AJ, Hengel SR, Rapchak KS, Mitchell D, Pressimone CA, Fagunloye AA, Luong TT, Russell RA, Vyas RK, Mertz TM, Zaher HS, Mosammaparast N, Malc EP, Mieczkowski PA, Roberts SA, Bernstein KA | 2021 | Raw sequencing reads in fastq format | http://www.ncbi.nlm.nih.gov/bioproject/?term=PRJNA694993 | NCBI BioProject, PRJNA694993 |

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
