## [Editor Report]

The study supports the existence of a new pathway for the removal of an important DNA lesion, 3meC in single stranded DNA during replication, that seems essential in yeast, but likely contributes in other organisms, and helps clarify the distinctive role of homologous recombination in DSB repair and post-replicative repair. The paper is of interest to an audience of DNA repair and cancer biologists because it seeks to refine the mechanism by which cells respond to DNA damage. This is an important topic, and the results are consistent with previous work in the field. New claims could change our understanding of DNA repair with implications for mutagenesis and cancer therapy.

---

## [Decision Letter]

**Decision letter after peer review:**

Thank you for submitting your article "The Shu Complex Prevents Mutagenesis and Cytotoxicity of Single-Strand Specific Alkylation Lesions" for consideration by *eLife*. Your article has been reviewed by 3 peer reviewers, one of whom is a member of our Board of Reviewing Editors, and the evaluation has been overseen by a Reviewing Editor and Jessica Tyler as the Senior Editor. The reviewers have opted to remain anonymous.

Essential revisions:

The Bernstein lab has made relevant contributions to understand the role of the Shu complex in HR-mediated repair. In this new study, the authors show that Shu is critical for 3meC damage tolerance in yeast and that expression of human ALKBH2, responsible for the repair of 3meC rescues the MMS-sensitivity of Shu mutants but not that of HR mutants. It is a provocative idea that the Shu complex may directly detect alkylated bases, such as 3-methylC, as well as various DNA repair intermediates. The study is relevant for the DDR field and merits publication, provided that the following suggestions or questions are attended.

1. Would the mutation rate of the double mutants csm2 mag1 show an increase compared to single mutants? These data can be added. Analysis of can1 resistant mutants would work.

2. The transcriptional asymmetry displaying higher densities of C mutations on the NT strand consistent with either TC-NER or preferential formation of 3meC in NT can be tested using rad26 csm2 double mutants. Analysis of mutations at the can1 gene should be sufficient. Alternatively, if authors have a system to turn transcription OFF and ON to study mutations, they can study mutations in both conditions.

3. A view of certain regions, like CAN1, showing all mutations, would be good to figure out clustering of mutations, if any, and some other parameters.

4. What's the effect of ALKB2 on the mutation rate in HR mutants in MMS? One at least could be tested.

5. A parallel study on the mutation rate (can-resistant mutations would work) in MMS-untreated and treated cells should be done for rad55 csm2 and rad55-SA csm2 mutants tested in (Figure 5). Could the MMS sensitivity of the rad55-SA allele be reproduced from the previous work?

6. While the Y2H is a good approach, co-IP is necessary to conclude the reduced interaction of Rad55-SA with Csm2.

7. It would strengthen the impact of the work if the mechanism could be extended to other types of DNA lesions. In other words, does Csm2-Psy3 also bind to other types of lesions in ssDNA/flapped DNA?

8. Line 296-298: The WT +/- ALKBH2 should be included in figure 4A to support this conclusion.

9. The mutation spectra is very similar in the presence and absence of csm2. Is this true? Did authors collect sufficient data with the WT strain to quantitatively compare mutation spectra? This would be interesting to address directly in the manuscript. This has implications for the function of csm2 if it were to be more important for specific types of mutation events.

10. What is the physiological significance of a double-flap structure? It would have helped to have some minimal discussion of how this type of structure might form. Is there greater specificity for binding of the Shu-complex to abasic sites or alkylated sites in a single flap that might mimic a stalled replication fork. It feels like a lot is made of this 2-fold difference in binding affinity and not enough attention to what might be present in the cell for the Shu complex to recognize. It is very surprising that there is a big anisotropy change with a labeled single strand oligo. Does this indicate that Shu is binding to the entire 5' flap, including the label? It is stated that the raw binding data is provided, but I could not find it in the supplemental tables. I would have liked to see how total fluorescence might change as a function of protein concentration. I also noticed that Kd values are different in each sequential paper. This could be due to improved protein quality, but it would be helpful to spend more attention to the quality of the protein. If the protein binds tightly, then it is beneficial to perform a binding assay under high concentration of DNA in order to determine the amount of protein capable of binding the DNA. This would be an excellent biochemical assay to compare different sources of protein.

---

## [Author Response]

Essential revisions:1. Would the mutation rate of the double mutants csm2 mag1 show an increase compared to single mutants? These data can be added. Analysis of can1 resistant mutants would work.

The mutation rate for csm2 mag1 double mutants was determined in Godin et al., NAR. The rate of MMS-induced mutation is increased in the csm2 mag1 double mutants compared to the csm2 and mag1 single mutants. In the revision to this manuscript, we generated MMS-induced *CAN1* mutation spectra for the csm2, mag1, and csm2 mag1 double mutants. This data has been added to Figure 1—figure supplement 3. We found that the csm2 mag1 double mutants have a mutation spectrum similar to the mag1 single mutants, but different from the csm2 single mutant. This is due to csm2 mag1 double mutants and mag1 double mutants having more substitutions at A:T base pairs, presumably due to the lack of repair mechanisms for 3meA. This data further supports a role for the Shu complex in the error-free bypass of 3meA.

2. The transcriptional asymmetry displaying higher densities of C mutations on the NT strand consistent with either TC-NER or preferential formation of 3meC in NT can be tested using rad26 csm2 double mutants. Analysis of mutations at the can1 gene should be sufficient. Alternatively, if authors have a system to turn transcription OFF and ON to study mutations, they can study mutations in both conditions.

Since a transcription induction system would be expected to increase both TC-NER and the formation of 3meC on the NT, we created a rad26 csm2 double mutant strain and measured *CAN1* mutation rates and spectra. This data has been added in Figure 2—figure supplement 1. It indicates that co-deletion of rad26 with csm2 does not significantly increase the *CAN1* mutation rate above that of csm2. Additionally, substitutions in *CAN1* of MMS-treated csm2-deficiency yeast favor C-based mutations in the NT over G-based mutations (which would be C in the transcribed strand). This slight transcriptional asymmetry was still present in the csm2 rad26 strain, suggesting that the transcriptional asymmetry of C mutations is due to the preferential formation of 3meC on the NT.

3. A view of certain regions, like CAN1, showing all mutations, would be good to figure out clustering of mutations, if any, and some other parameters.

Due to the selection bias that would occur for mutations in CAN1 that were obtained by selecting for canavanine-resistance, we decided to create a graphical representation of non-selected MMS-induced mutations in whole genome sequencing of WT and csm2-deficient yeast on chromosome V. This graphic nicely shows the increased density of mutations in the csm2Δ isolates compared to WT yeast. However, no obvious clustering was observed. This data is now in Figure 1—figure supplement 2.

4. What's the effect of ALKB2 on the mutation rate in HR mutants in MMS? One at least could be tested.

We thank the reviewers for this valuable suggestion. We have tested the effect of ALKBH2 on the mutation rate in *rad51∆* cells (New Figure 5C). As predicted by our model, the mutation rate of MMS-treated *rad51∆* cells is significantly reduced when ALKBH2 is expressed but not with the catalytically inactive ALKBH2 mutant.

5. A parallel study on the mutation rate (can-resistant mutations would work) in MMS-untreated and treated cells should be done for rad55 csm2 and rad55-SA csm2 mutants tested in (Figure 5). Could the MMS sensitivity of the rad55-SA allele be reproduced from the previous work?

We thank the reviewers for this suggestion. We have now included the results of a *CAN1* mutagenesis assay for untreated and MMS-treated *rad55∆ csm2∆* and *rad55-SA csm2∆* mutant cells (New Figure 5F). Consistent with our MMS sensitivity assay, the *rad55-SA csm2∆* double mutant cells showed no increased spontaneous or MMS-induced mutation rates compared to the *csm2∆* single mutant cells.

As the reviewer noted, the MMS sensitivity of the rad55-SA allele is reproducible from prior work from the Heyer lab (Hertzberg et al., (2006) MCB 26(22): 8396; Figure 3A). Although we observe less sensitivity at 0.02% relative to the 0.03% (Figure 5E). This small difference could be due to the Heyer lab incubating their plates for 8 days at 20°C whereas we grew ours at 30°C for 48 hours. Overall, the findings are the same and reproducible.

6. While the Y2H is a good approach, co-IP is necessary to conclude the reduced interaction of Rad55-SA with Csm2.

We agree with the reviewers that a co-IP would be a wonderful complementary approach to the Y2H. However, the interaction between Rad55 and Csm2 has been shown by Y2H and direct in vitro pull-downs, and we have failed to detect it by co-IP despite several attempts by multiple individuals. We have tried by pulling down on either Rad55 or Csm2, as well as treating the cells with MMS and using S-phase synchronized cells as well. It is possible that this interaction, despite its importance, is too transient or weak to be detected by co-IP. We have now addressed this point in the Results section.

7. It would strengthen the impact of the work if the mechanism could be extended to other types of DNA lesions. In other words, does Csm2-Psy3 also bind to other types of lesions in ssDNA/flapped DNA?

We agree with the reviewer that it is important to examine multiple substrates with a myriad of MMS-induced lesions. We have had substantial delays in receiving modified oligos due to COVID-19 and supply chain issues from the vendors. For example, it took 7 months to get the one 3meC substrate and we were unable to examine this modification at other locations in the DNA. Similarly, it took 4 months to receive the 1meA substrate. We have now performed the equilibrium titrations with both 3meC and 1meA and while we observe binding, we do not observe an increase in binding affinity. More details regarding these findings are addressed in the points below.

8. Line 296-298: The WT +/- ALKBH2 should be included in figure 4A to support this conclusion.

We have now included data from MMS-treated WT cells with and without ALKBH2 (New Figure 4A, lanes 1-3). As expected, we observe a slight but not significant difference between the mutation rates of WT with or without ALKBH2.

9. The mutation spectra is very similar in the presence and absence of csm2. Is this true? Did authors collect sufficient data with the WT strain to quantitatively compare mutation spectra? This would be interesting to address directly in the manuscript. This has implications for the function of csm2 if it were to be more important for specific types of mutation events.

We identified ~2400 MMS-induced mutations in WT yeast and ~8800 MMS-induced mutations in csm2-deficient yeast. These numbers of mutations are sufficient to quantitatively compare the mutation spectra. It is true that the WT and csm2 spectra are very similar, except that csm2-deficient yeasts have many more mutations. This is likely due to A:T mutations in WT and csm2-deficient yeast being both caused by 3meA and C:G mutations being caused by a possible combination of both 3meC and 7meG. 7meG and 3meC can produce similar mutation substitution patterns. The role of the Shu complex in bypassing multiple MMS-induced lesions is now discussed in the discussion of the manuscript.

10. What is the physiological significance of a double-flap structure? It would have helped to have some minimal discussion of how this type of structure might form. Is there greater specificity for binding of the Shu-complex to abasic sites or alkylated sites in a single flap that might mimic a stalled replication fork. It feels like a lot is made of this 2-fold difference in binding affinity and not enough attention to what might be present in the cell for the Shu complex to recognize.

We thank the reviewer for pointing this out and have now clarified this point. When a replication fork stalls, the replicative helicase, and polymerase can uncouple and ssDNA can form behind the fork on the lagging strand. Our previous studies in Rosenbaum et al., 2019 showed that the DNA binding subunits Csm2-Psy3 bind similarly to double-flap, 5’ flap, 3’ flap, and static flaps (Rosenbaum et al., (2019) Nature Communications, 3515; Supplementary Figure 5B). Our in vivo data supports the model that the Shu complex functions preferentially on the lagging strand and to enable error-free bypass of multiple MMS-induced DNA lesions in this context. We have now addressed these points in the discussion.

It is very surprising that there is a big anisotropy change with a labeled single strand oligo. Does this indicate that Shu is binding to the entire 5' flap, including the label?

While there is a substantial change in anisotropy, this assay does not provide information regarding where the Shu complex binds on this oligonucleotide. We have tried to determine stoichiometry using several different approaches. We tried FAM fluorescence and anisotropy under stoichiometric conditions (saturating DNA concentrations, 1 µM). We have also tried atomic force microscopy and structural studies but have not been successful to date.

It is stated that the raw binding data is provided, but I could not find it in the supplemental tables. I would have liked to see how total fluorescence might change as a function of protein concentration.

We apologize for this oversight; the anisotropy files have been added.

I also noticed that Kd values are different in each sequential paper. This could be due to improved protein quality, but it would be helpful to spend more attention to the quality of the protein. If the protein binds tightly, then it is beneficial to perform a binding assay under high concentration of DNA in order to determine the amount of protein capable of binding the DNA. This would be an excellent biochemical assay to compare different sources of protein.

Unfortunately, our initial studies in the first draft of this manuscript were performed with a protein preparation that aggregated and thus we thought that the binding affinity was increased due to increased protein purity. Our updated purification protocol has been added in the manuscript in the Materials and methods. We fixed this aggregation by modulating our purification scheme, performed binding assays with three preparations of recombinant Csm2-Psy3, and now observe binding isotherms consistent with our previous publications (Rosenbaum et al., 2019).

We agree with the reviewer that understanding the stoichiometry is important for understanding how Csm2-Psy3 binds to double-flap substrates. We have tried to perform stoichiometric titrations with higher concentrations of DNA (400 nM and 1 uM) but the protein at these salt concentrations aggregates and this, unfortunately, prohibits obtaining stoichiometries.